

# Supervised learning of few dirty bosons
# with variable particle number

**Pere Mujal[1,2,3]⋆, Àlex Martínez Miguel[1,2], Artur Polls[1,2],**
**Bruno Juliá-Díaz[1,2] and Sebastiano Pilati[4]**

**1** Departament de Física Quàntica i Astrofísica, Universitat de Barcelona,
Martí i Franquès 1, 08028 Barcelona, Spain
**2** Institut de Ciències del Cosmos (ICCUB), Universitat de Barcelona,
Martí i Franquès 1, 08028 Barcelona, Spain
**3** Institut de Física Interdisciplinària i Sistemes Complexos (IFISC),
Universitat de les Illes Balears - CSIC, UIB Campus, 07122 Palma, Mallorca, Spain
**4** School of Science and Technology, Physics Division, Università di Camerino,
62032 Camerino (MC), Italy

⋆ peremujal@gmail.com

## Abstract

We investigate the supervised machine learning of few interacting bosons in optical speckle disorder via artificial neural networks. The learning curve shows an approximately universal power-law scaling for different particle numbers and for different interaction strengths. We introduce a network architecture that can be trained and tested on heterogeneous datasets including different particle numbers. This network provides accurate predictions for all system sizes included in the training set and, by design, is suitable to attempt extrapolations to (computationally challenging) larger sizes. Notably, a novel transfer-learning strategy is implemented, whereby the learning of the larger systems is substantially accelerated and made consistently accurate by including in the training set many small-size instances.



# 1  Introduction

Supervised machine learning is emerging as a potentially disruptive technique to accurately predict the properties of complex quantum systems. It has already allowed researchers to drastically speedup various important computational tasks in quantum chemistry and in condensed-matter physics [1, 2], including: molecular dynamics simulations [3–7], electronic structure calculations [8–13], structure-based molecular design [14–16], and protein-molecule binding-affinity predictions [17–19]. Deep neural networks represent the most powerful and versatile models. In principle, they can approximate any continuous function with arbitrary accuracy [20]. However, training them without overfitting requires extremely copious datasets, often comprising hundreds of thousands of training instances. Generating such datasets for large quantum systems is computationally impractical, unless one accepts (sometimes unreliable) approximations such as, e.g., density functional theory. This represents a critical problem that hampers the further development of machine-learning techniques for quantum systems.

A possible approach to circumvent the above problem is to adopt a transfer-learning strategy, as often done in the field of image analysis [21]. In the case of quantum systems, transfer learning can be implemented by scaling to larger sizes the neural networks that have been trained on smaller – therefore, computationally tractable – systems. In fact, a form of size scalability is currently being employed in the field of molecular dynamics simulations; in that approach, the ground-state energies are computed as the sum of single-atom contributions, but taking into account only the short-range atomic environments (see, e.g., Ref. [22]). Proper scalability has recently been implemented in a few distinct ways: i) assuming the extensivity property, using properly constructed size-extensive networks [23]; ii) adopting normalized descriptor-vectors of fixed size (i.e., independent of the physical system size) [24]; iii) implementing scalable convolutional networks via global pooling layers, for systems of variable spatial extent [25]. To the best of our knowledge, statistical models that accept the particle number as an explicit system descriptor have not been investigated yet.

In this article, we consider the supervised learning of interacting bosons in a one-dimensional random external field. Our main goals are to quantify the learning speed [26], in terms of prediction accuracy versus number of instances in the training set, and to implement flexible neural networks that can address different particle numbers simultaneously. The Hamiltonian we focus on is realistic and describes experiments performed with ultracold atoms in optical speckle fields [27]. It represents a challenging computational task, belonging to the family of dirty boson problems [28–30]. Recently, this model has been addressed in a study on the stability of the Anderson localization phenomenon against inter-particle interaction [31]. The model was shown to host a many-body localized phase. Here, we analyse how many instances are needed to train deep neural networks to accurately predict its ground-state energy, depending on the interaction strength and on the particle number. The training and the test

sets are produced via an exact diagonalization technique. This choice allows us to avoid the common approximations employed in most supervised machine-learning studies. However, it limits our analysis to small particle numbers, specifically, up to four bosons.

Notably, we implement a neural network for continuous-space quantum systems with variable particle number. This network combines the scalable convolutional architecture of Ref. [25] [see Fig. 1, panel (b)], which can address disordered systems of variable spatial extent (but fixed particle number), with an additional descriptor representing the particle number. This descriptor bypasses the convolutional layers and is fed directly to the final dense layers [see Fig. 1, panel (a)]. As we demonstrate here, this network is able to accurately predict the ground-state energies of systems with different particle numbers, even when considering heterogeneous datasets including instances with different size. The learning speed appears to be independent of the particle number and of the interaction strength. In fact, the prediction accuracy follows an approximately universal power-law scaling with the number of instances in the training set. Our neural network can also be used to attempt extrapolations to particle numbers larger than those included in the training set. The extrapolation accuracy depends on the interaction strength, and it improves if the training set is copious and includes (relatively) large particle numbers. Furthermore, we show that the learning of the larger sizes can be substantially accelerated if a training set with many small-size instances, which are computationally accessible, is merged with a small amount of instances for the larger particle number. This strategy provides consistently accurate predictions, also for the larger size, with a computationally feasible training set. It represents an alternative transfer-learning technique, paving the way to a novel approach to accurately predict the properties of complex quantum systems, for which copious training sets cannot be generated in feasible computational times.

Our study begins with an analysis of the learning speed, considering both datasets with a unique particle number, as well as the combined learning with heterogeneous datasets. Then, we analyse the accuracy of the extrapolations to particle numbers larger than those included in the training set, as well as the accelerated learning of relatively large systems using data for smaller sizes. In detail, the rest of the article is organized as follows: the physical system we address and the computational method we employ to determine its ground-state energy are provided in Section 2, together with a description of the artificial neural network introduced in this article and some details on the training algorithm. The analysis on the learning speed of the few-boson problem is reported in Section 3. Section 4 reports the analysis on the extrapolation procedure and on the accelerated learning. The summary of our main findings and some future perspectives are reported in Section 5.

## 2 Model and Methods

### 2.1 Physical system: few 1D dirty bosons

We consider a one-dimensional system of few repulsively interacting bosons in the presence of an external disordered potential. This model has been experimentally engineered and describes ultracold atoms subjected to optical speckle fields and confined in cigar-shaped traps. Specifically, it corresponds to the setup of early cold-atom experiments on the Anderson localization phenomenon [32,33]. The Hamiltonian of the system reads

$$\mathcal{H} = \sum_{i=1}^{N} \left( -\frac{\hbar^2}{2m} \frac{\partial^2}{\partial x_i^2} + V(x_i) \right) + \sum_{i<j}^{N} \nu(x_i, x_j), \tag{1}$$

where $m$ is the particle mass, $N$ the number of particles, and $x_i$ corresponds to the position of particle $i$, with $i = 1, \ldots, N$. The two-body interaction is described by a contact interaction

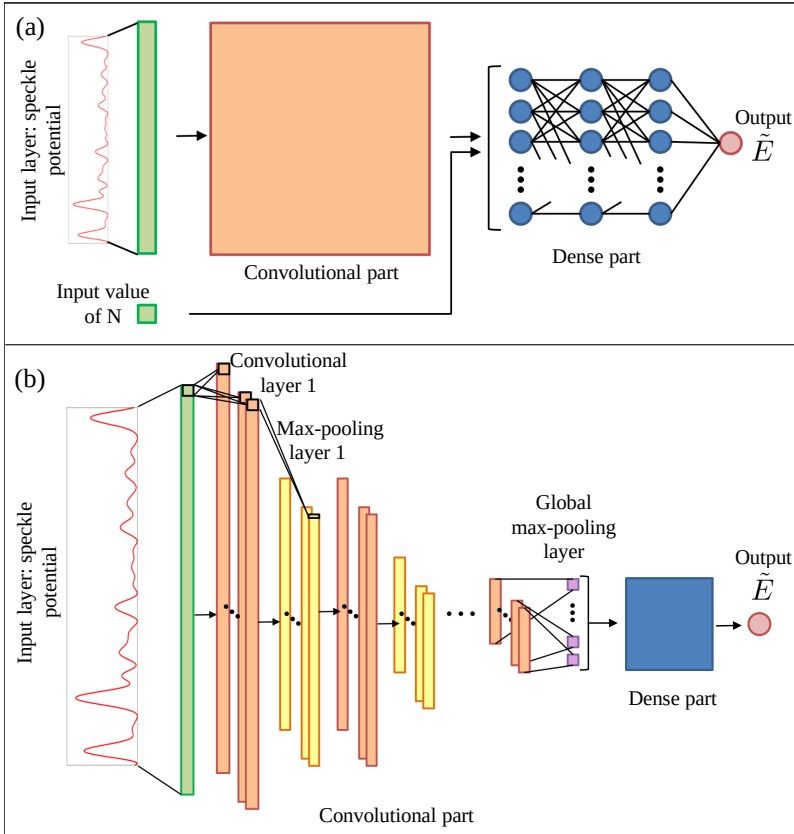

Figure 1: (a) Schematic representation of the deep feed-forward neural network used to predict the ground-state energy, $\tilde{E}$, of few-boson systems (output). The input descriptors are the values of the speckle potential on a fine discrete grid. In the case of training with heterogeneous datasets, an additional system descriptor is included, representing the particle number $N$. This descriptor is connected directly to the dense part of the network. (b) Structure of the convolutional part of the neural network. This model is used when training on homogeneous datasets including instances with a unique particle number.

potential,

$$v(x_i, x_j) = g\,\delta(|x_i - x_j|), \tag{2}$$

where $g$ is the parameter that defines the interaction strength. Its sign determines the character of the interaction: repulsive for $g > 0$ and attractive for $g < 0$. We focus on the repulsive case.

The external potential $V(x)$ represents the effect of optical speckle fields on ultracold atoms. It can be generated on a discrete spatial grid with fine spacing via the stochastic numerical algorithm described in detail in Refs. [34, 35]. We produce many instances of speckle potentials using different pseudo-random numbers. All instances are characterized by the same spatial correlation length, indicated in the following as $\ell$, and by the same average intensity $V_0$. The correlation length allows one to define a characteristic energy scale, namely, the correlation energy $E_c = \hbar^2/(m\ell^2)$. In the following, we consider speckle fields of fixed spatial extent, namely, $L = 20\ell$, with hard-wall boundary conditions. The spatial grid for the speckle potential includes 1024 points, corresponding to a grid spacing $\delta x \simeq 0.153\ell$. With such a fine grid, discretization effects are negligible. The disorder strength is fixed at $V_0 = 5E_c$. Different values of the interaction parameter $g$ are considered; they are expressed in the following in units of $\hbar^2/(\ell m)$. Specifically, from now on we consider the weak interaction $g = 0.05$, an

Table 1: Details of the layers constituting the convolutional and the dense parts of our neural network. Definitions are standard, see for instance Ref. [39].

| Layer name | Layer Function | Layer description |
|---|---|---|
| Input layer 1 | Input | The value of the speckle potential in 1024 points |
| Convolutional layers | ReLu | 50 filters, kernel_size=5, strides=1, padding=same, activation=relu |
| Local max-pooling layers | Max-pooling | Local pooling with pool_size=3 |
| Global max-pooling layer | Global max-pooling | 50 neurons = number of filters in the preceding convolutional layer |
| Input layer 2 | Input | $N$, number of particles |
| Dense layers | ReLu | 30 neurons, activation=relu |
| Output layer | Identity | 1 neuron, activation=identity |

intermediate value $g = 0.26$, and the strong-coupling case $g = 1$.

We train deep neural networks to predict the ground-state energies of different instances of the Hamiltonian (1). These energies are computed by means of the exact-diagonalization method described in Ref. [31]. This method is based on a second-quantization formalism. The Fock space of the $N$ bosons is built using the basis of the single-particle eigenstates of the kinetic energy operator. The diagonalization is performed in a truncated space including only the Fock basis states with kinetic energy smaller than a chosen threshold, following the technique introduced in Ref. [36]. This energy threshold determines both the dimension of the truncated $N$-boson Fock space $D_{MB}$, and the required number of single-particle basis states $M$. Further details on the computational technique we employ are reported in Ref. [37]. The energy thresholds we adopt in this article lead to the following truncation parameters: for $N = 1$, we have $M = D_{MB} = 100$; for $N = 2$, $M = 100$ and $D_{MB} = 3914$; for $N = 3$, $M = 100$ and $D_{MB} = 88106$; and for $N = 4$, $M = 80$ and $D_{MB} = 552099$. The computational resources available to us allow producing datasets including different numbers of instances; specifically, we produce 600000, 50000, 2000, and 270 instances for $N = 1, 2, 3$, and 4, respectively, for each of the three values of the interaction parameter $g$ we consider. These datasets are available at [38].

## 2.2 Network architecture

In Ref. [25], deep feed-forward neural networks have been employed in the supervised learning of the ground-state energy of the Hamiltonian (1). However, that study addressed only the single particle case, namely, the case $N = 1$. A scalable architecture was implemented using standard convolutional layers connected to dense hidden layers (i.e., with all-to-all connectivity) via a global pooling operation. This allows the model to address disordered systems of arbitrary spatial extent $L$. Our goal is to further develop that architecture so that it can address also an arbitrary particle number $N$.

Our investigation first addresses homogeneous datasets including instances with a single particle number, either $N = 1$, $N = 2$, or $N = 3$. For this purpose, the architecture of Ref. [25] (represented in panel (b) of Fig. 1) can be employed without modifications. The system instances are represented by 1024 descriptors corresponding to the speckle potential intensities $V(x_k)$ on the spatial grid $x_k = k\delta x$, with $k = 0, ..., 1023$. Since the grid spacing $\delta x$ is much smaller than the disorder correlation length $\ell$, these 1024 descriptors provide an exhaustive representation of the speckle potential of each instance. The 1024 descriptors are fed to the convolutional part of the architecture. This part includes six convolutional layers with 50 fil-

ters, each followed by a local pooling layer. The output of the convolutional part is forwarded to the first of three dense layers, each including 30 neurons, via a global pooling layer. The final layer includes a single neuron. Its activation should correspond to the ground-state energy. Thanks to the global pooling layer, this architecture can be applied to systems with different spatial extent $L$ (and, hence, different numbers of descriptors), without re-training.

To address heterogeneous datasets containing instances with different particle numbers, we have to extend the architecture shown in panel (b) of Fig. 1. Specifically, we include an additional descriptor whose value corresponds to the particle number $N$. The corresponding neuron is linked directly to the first dense layer, bypassing the convolutional and the pooling layers (see panel (a) of Fig. 1). In principle, this should allow the model to learn how the ground-state energy depends on the particle number, providing predictions for arbitrary $N$. In Sections 3 and 4 we quantify if and to what extent this goal is achieved. All details of the neural-network structure are reported in Table 1.

### 2.3 Training procedure

The training is performed by minimizing the mean squared error (MSE), defined as:

$$\text{MSE} = \frac{1}{N_{\text{train}}} \sum_{t=1}^{N_{\text{train}}} \left( \tilde{E}_t - E_t \right)^2 , \tag{3}$$

where $E_t$ and $\tilde{E}_t$ are, respectively, the exact and the predicted ground-state energies of the training instance $t$. $N_{\text{train}}$ is the number of instances included in the training set. The optimization of the neural-network weights and biases is performed using the *Adam* algorithm [40], as implemented in the *Keras* python library [41]. An early stopping criterion is adopted. It is based on the MSE of a validation set (distinct from the training set). The optimal network parameters obtained throughout the training process are retained. To quantify the accuracy of the predictions provided by the trained networks we consider two figures of merit. The first is the mean absolute error (MAE), defined as:

$$\text{MAE} = \frac{1}{N_{\text{test}}} \sum_{t=1}^{N_{\text{test}}} \left| \tilde{E}_t - E_t \right| . \tag{4}$$

The second is the coefficient of determination, defined as:

$$\text{R}^2 = 1 - \frac{\sum_{t=1}^{N_{\text{test}}} \left( \tilde{E}_t - E_t \right)^2}{\sum_{t=1}^{N_{\text{test}}} \left( E_t - \langle E \rangle \right)^2} . \tag{5}$$

Here, $N_{\text{test}}$ is the number of instances in the test set, and $\langle E \rangle$ is their average ground-state energy. We stress that the instances included in the test set are distinct from those used for training and for validation. It is worth recalling that perfect predictions correspond to the score $R^2 = 1$, while a constant function predicting the correct average $\langle E \rangle$ corresponds to the score $R^2 = 0$. For the results reported in the following sections, unless otherwise specified, 20% of the datasets are used for testing. The remaining 80% is divided into the training data, accounting a 75%, and validation data, corresponding to the remaining 25%. It is worth mentioning that the data reported in this article are obtained without regularization techniques. In our analysis, we inspect for the occurrence of overfitting by monitoring the discrepancy between the MAE of the training set and of the validation set. In general, we find similar results, apart for the smallest training sets, for which the validation MAE is, in the worst case, up to twice the training MAE. Anyway, tuning the regularization parameter with a standard L2 regularization [39] does not significantly improve the accuracy on the test set.

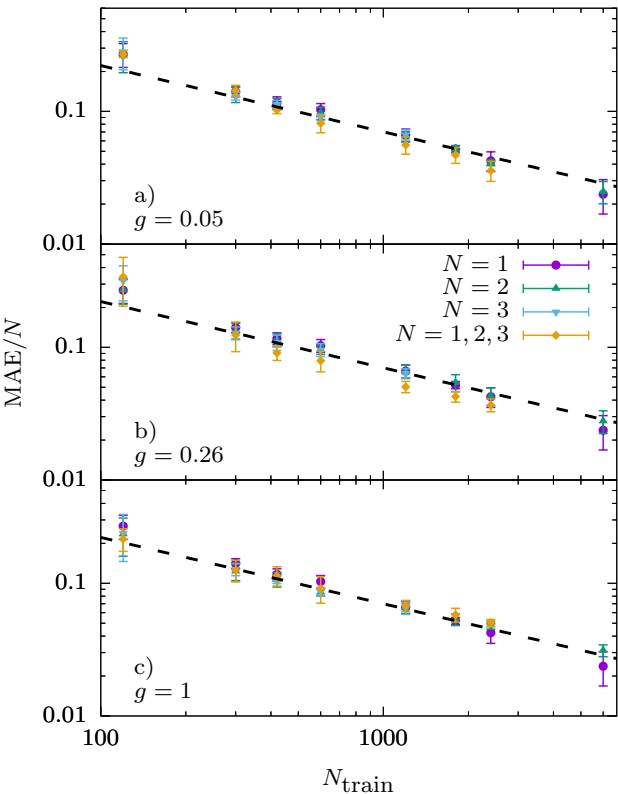

Figure 2: Mean absolute error per particle MAE/$N$, computed on the test ground-state energies, as a function of the number of instances in the training set $N_{\text{train}}$. The different symbols correspond to training on homogeneous datasets including a unique particle number (either $N = 1$, $N = 2$, or $N = 3$) and to combined training and testing on heterogeneous datasets including all three particle numbers ($N = 1, 2, 3$). The three panels correspond to different interaction strengths $g$. The errorbar is the estimated standard deviation of the mean obtained with up to eight independent models trained with different pseudo-random numbers. The dashed line corresponds to a power-law scaling with $b = 0.5$, see text for details.

## 3  Learning the few-body problem

### 3.1  Homogeneous datasets

The neural networks described in the previous section are trained to predict the ground-state energy of the Hamiltonian (1). We first analyse homogeneous datasets including instances with a unique particle number. In this first analysis, the networks are trained and tested on the same system size, considering the cases $N = 1$, $N = 2$, and $N = 3$ separately. Since in this analysis the particle number is fixed, we adopt the network architecture shown in panel (b) of Fig. 1, i.e., the one that only accepts external potential values as system descriptors. Three values of the interaction parameter are (separately) considered, namely, $g = 0.05$, $g = 0.26$, and $g = 1$. The first choice corresponds to the weakly-interacting regime, where the ground-state energies are not far from their non-interacting values. The second choice represents an intermediate interaction strength, and the third choice is close to the Tonks-Girardeau limit where the bosonic ground-state energy approaches the result corresponding to (non-interacting) identical fermions. The learning speed is analysed in Fig. 2. The pre-

diction accuracy, as measured by the MAE per particle computed on the test set, is plotted as a function of the number of instances included in the training set $N_{\text{train}}$. It is worth reminding that the test is performed on instances not included in the training and in the validation sets. In general, one expects a power-law scaling of the prediction accuracy, corresponding to $\text{MAE}/N \propto N_{\text{train}}^{-b}$ [42, 43], where $b > 0$. Interestingly, we find that the data for all particle numbers and for all interaction strengths we consider are consistent with a power-law scaling with the same exponent $b = 0.5$ (see dashed line in Fig. 2). These results suggest an approximate universal behavior, at least for the one-dimensional many-body localized model we address. While our focus is on the scaling exponent $b$, one notices that the datasets corresponding to different $N$ and $g$ essentially overlap, within the statistical uncertainties. This suggests that also the prefactor is, at least approximately, universal.

## 3.2 Heterogeneous datasets

One of our main goals is to implement models that can address different system sizes simultaneously. This is achieved via the modified neural-network shown in panel (a) of Fig. 1. This model is fed with an additional descriptor representing the particle number $N$, beyond the 1024 speckle potential intensities. We train and test this model using heterogeneous datasets which include system instances with different particle numbers, with equal populations for the three $N$ values. As before, training and testing are performed for the same interaction parameter, addressing separately the three values we consider. We stress, however, that in this case the same neural network predicts ground-state energies for different particle numbers, while in the previous analysis different models were employed for each case. Notably, the MAE per particle follows the same power-law scaling with exponent $b = 0.5$, as previously found in the analysis with separate particle numbers. This further supports the statement about an approximately universal behaviour.

# 4 Extrapolation and accelerated learning

The computational cost required to solve many-body problems increases exponentially fast with the number of particles. For example, with our exact-diagonalization technique the cost increases by a factor $\approx 27$ going from the $N = 2$ case to the $N = 3$ case, as well as when going from the $N = 3$ to the $N = 4$ case. Hence, one expects that the datasets one encounters in practical scenarios contain many small $N$ instances, and only very few instances for relatively large $N$. It is therefore natural to wonder (i) if a variable-$N$ neural network can perform extrapolations, providing predictions for system sizes larger than those included in the training sets, and (ii) if the many small-$N$ instances can be used to accelerate the training process for larger $N$, enabling the network to provide accurate predictions even when only very few training instances are available for the larger system size. In the following, we address these relevant issues using the variable-$N$ architecture shown in panel (a) of Fig. 1. First, in Section 4.1 we focus on the extrapolation and on the accelerated learning of the $N = 3$ case, using data for $N = 1$ and $N = 2$; then, in Section 4.2 we address the $N = 4$ case, where we use data for $N = 1, 2$, and 3. Finally, in Section 4.3 we consider a real-case scenario with much larger databases for lower particle numbers.

## 4.1 Extrapolation and accelerated learning for three particles

In the first case, a network is trained on a dataset including 1800 instances for $N = 1$ and as many for $N = 2$. This network is then used to predict the ground-state energies of $N = 3$ instances. To quantify the prediction accuracy we consider the MAE and the coefficient of

Table 2: Performance of the neural network in the test-case considered in Figs. 3 and 4. The coefficient of determination $R^2$ and the mean absolute error MAE are reported for three interaction strengths $g$, considering networks trained on $N = 1, 2$ and on $N = 1, 2, 3$. The test results are shown for the particle numbers $N$ included in the training set (number of training instances in parenthesis), for the extrapolations to $N = 3$ and to $N = 4$, and for the accelerated learning with additional large-size instances in the training sets.

| | $g = 0.05$ | | $g = 0.26$ | | $g = 1$ | |
|---|---|---|---|---|---|---|
| | $R^2$ | MAE | $R^2$ | MAE | $R^2$ | MAE |
| Trained with $N = 1, 2$ | | | | | | |
| $N = 1$ (1800) | 0.992 | 0.027 | 0.991 | 0.027 | 0.987 | 0.035 |
| $N = 2$ (1800) | 0.994 | 0.047 | 0.995 | 0.042 | 0.988 | 0.065 |
| $N = 3$ (Extrap.) | 0.912 | 0.299 | 0.880 | 0.366 | 0.848 | 0.374 |
| Accelerated learning for $N = 3$ | | | | | | |
| $N = 1$ (1800) | 0.993 | 0.024 | 0.991 | 0.029 | 0.987 | 0.037 |
| $N = 2$ (1800) | 0.992 | 0.046 | 0.994 | 0.045 | 0.991 | 0.057 |
| $N = 3$ (200) | 0.992 | 0.076 | 0.993 | 0.080 | 0.984 | 0.120 |
| Trained with $N = 1, 2, 3$ | | | | | | |
| $N = 1$ (1200) | 0.993 | 0.026 | 0.987 | 0.039 | 0.980 | 0.042 |
| $N = 2$ (1200) | 0.991 | 0.050 | 0.991 | 0.058 | 0.992 | 0.054 |
| $N = 3$ (1200) | 0.995 | 0.065 | 0.995 | 0.070 | 0.995 | 0.065 |
| $N = 4$ (Extrap.) | 0.977 | 0.172 | 0.920 | 0.313 | 0.830 | 0.481 |
| Accelerated learning for $N = 4$ | | | | | | |
| $N = 1$ (1200) | 0.987 | 0.037 | 0.981 | 0.046 | 0.980 | 0.049 |
| $N = 2$ (1200) | 0.987 | 0.068 | 0.988 | 0.068 | 0.987 | 0.069 |
| $N = 3$ (1200) | 0.991 | 0.093 | 0.990 | 0.099 | 0.990 | 0.099 |
| $N = 4$ (200) | 0.983 | 0.160 | 0.984 | 0.148 | 0.988 | 0.124 |

determination $R^2$. The corresponding values are reported in Table 2. For the system sizes included in the training set, namely, $N = 1$ and $N = 2$, the predictions are extremely accurate, corresponding to $R^2 \gtrsim 0.99$. The high degree of accuracy can be appreciated also in the scatter plots of Fig. 3 (panels (a), (c), and (e)), where the predicted energies for the test set are plotted as a function of the exact-diagonalization results. Interestingly, also the extrapolation to $N = 3$ are reasonably accurate, providing coefficients of determination $R^2 \gtrsim 0.85$ for all interaction strengths. The predictions appear to deviate from the exact values mostly in the large energy regime (see panels (b), (d), and (f) of Fig. 3). The distributions of absolute errors are shown in panels (b2), (d2), and (f2). The MAE per particle is around MAE/$N \simeq 0.1$. While this accuracy is remarkable, given that no $N = 3$ instance is exploited in the training process, it might not be sufficient for practical applications of supervised machine learning. Hence, we analyse the effect of adding to the previous training set just 200 instances for the particle number $N = 3$. We emphasize that the training is performed from scratch using the merged dataset. Interestingly, the combined training with the $N = 1$, $N = 2$, and $N = 3$ instances leads to high accuracy for all three system sizes. The coefficient of determination is $R^2 \gtrsim 0.99$. The MAE per particle for $N = 3$ is MAE/$N \simeq 0.03$, i.e., close to the accuracy obtained for $N = 1$ and for $N = 2$. Notably, the performances on the two smaller system sizes do not degrade. For the sake of comparison, it is worth noticing that, when the network is trained using only $N = 3$ instances (see Section 3), the MAE per particle reached with just 200 training instances is approximately an order of magnitude larger. These results indicate that the combined training with smaller

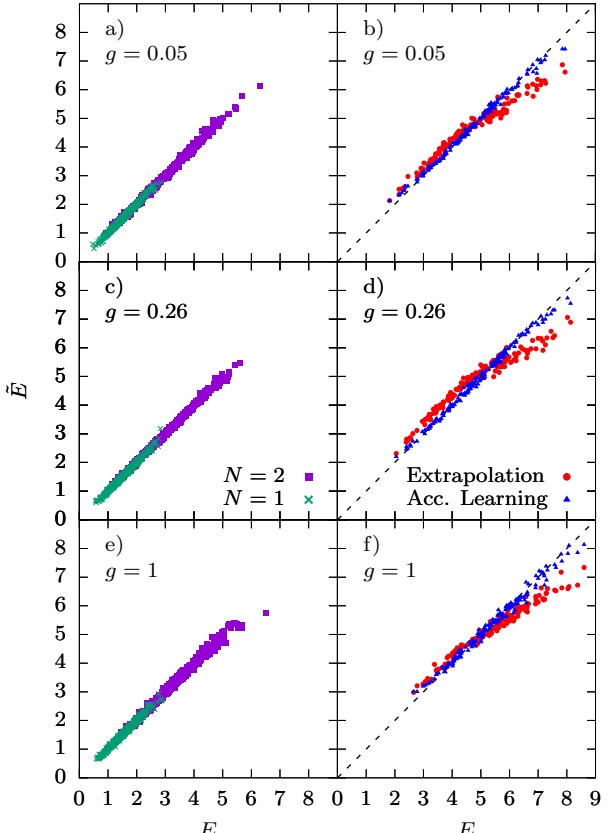
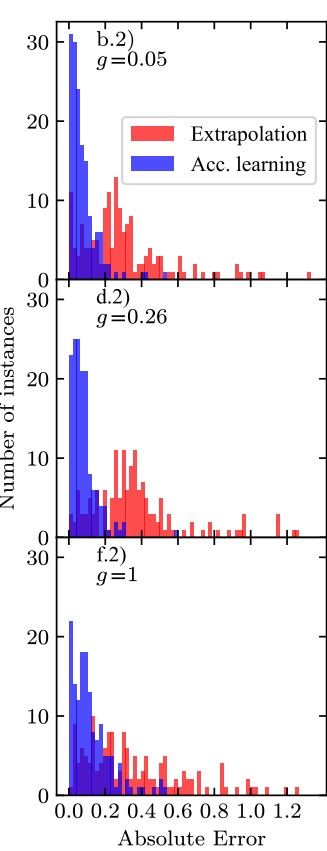

Figure 3: (Left) Ground-state energies $\tilde{E}$, predicted by a neural network trained on a heterogeneous dataset, as a function of the exact-diagonalization $E$. The training sets include 1800 instances for $N = 1$ and as many for $N = 2$. Panels (a), (c), and (e) report results for the systems sizes included in the training set. Panels (b), (d), and (f) report the extrapolations to the $N = 3$ case, and the accelerated learning with 200 additional instances for $N = 3$. (Right) Panels (b.2), (d.2), and (f.2) show the distributions of the absolute error, $|\tilde{E} - E|$, for the extrapolations and the accelerated-learning procedure corresponding to panels (b), (d), and (f), respectively. The three rows correspond to different interaction strengths $g$.

sizes provides a boost to the learning process for the larger size, allowing the network to reach high accuracy with fewer training instances.

## 4.2 Extrapolation and accelerated learning for four particles

The procedure described above is now extended to $N = 4$ systems. First, a network trained on a dataset including 1200 instances for $N = 1$, as many for $N = 2$ as well as for $N = 3$ (corresponding to a total of 3600 instances), is used to predict the ground-state energies of $N = 4$ instances. The scatter plots of these extrapolations are shown in Fig. 4. In the weak and intermediate interaction regime, the prediction accuracy is remarkably good, if one considers that no $N = 4$ instance is used in the training process. Specifically, the coefficient of determinations are: $R^2 \simeq 0.97$ for the weakly interacting case $g = 0.05$, $R^2 \simeq 0.92$ for $g = 0.26$, and

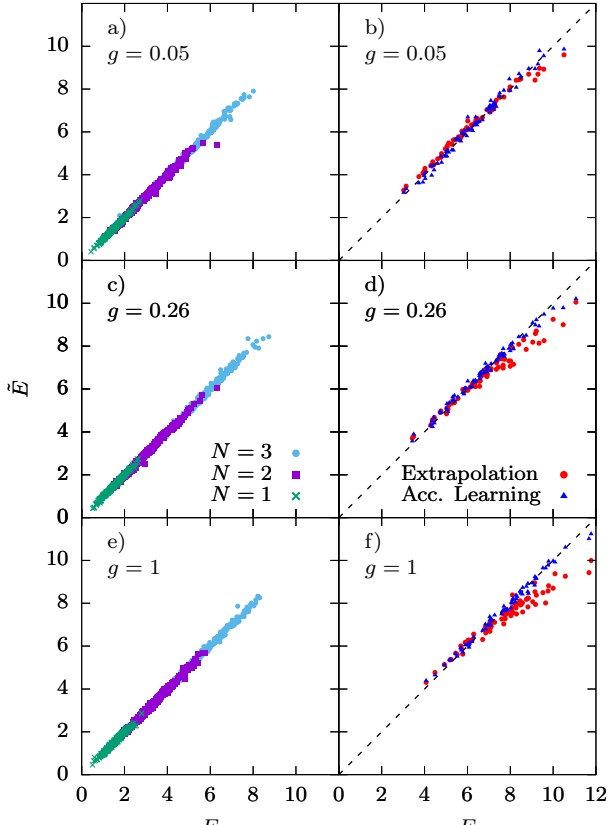
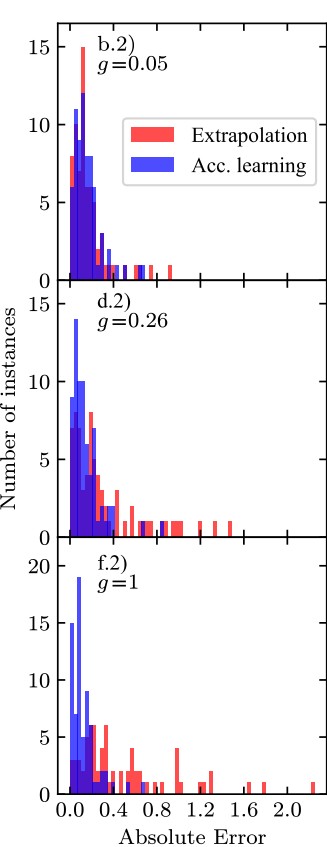

Figure 4: (Left) Ground-state energies $\tilde{E}$, predicted by neural networks trained on heterogeneous datasets, as a function of the exact-diagonalization results $E$. The training sets include 1200 instances for the particle numbers $N = 1, 2$, and 3, for a total of 3600 instances. Panels (a), (c), and (e) report results for the systems sizes included in the training set. Panels (b), (d), and (f) report the extrapolations to the $N = 4$ case, and the accelerated learning with 200 additional training instances for $N = 4$. (Right) Panels (b.2), (d.2), and (f.2) show the distributions of the absolute error, $|\tilde{E} - E|$, for the extrapolations and the accelerated-learning procedure corresponding to panels (b), (d), and (f), respectively. The three rows correspond to different interaction strengths $g$.

$R^2 \simeq 0.84$ for the strongly interacting case $g = 1$. This suggests that the network is learning how the ground-state energy scales with the particle number, at least for the weak and the intermediate interactions. However, the prediction accuracy is not always satisfactory. Next, we test the efficiency of accelerated learning. We include in the previous training set 200 instances for $N = 4$. As in Section 4.1, we find consistently accurate results, corresponding to a coefficient of determination $R^2 \gtrsim 0.98$ and a MAE$/N \sim 0.035$ for all interaction strengths. For comparison, a network trained only on 200 $N = 4$ instances (using the model of panel (b) of Fig. 1) would reach MAE$/N \sim 0.17$ ($R^2 \simeq 0.7$). Again, this indicates that transfer learning from smaller to larger system sizes is effective, allowing one to accelerate the training process for the larger systems.

Table 3: Performance of the neural network in the real-case scenario. The coefficient of determination $R^2$ and the mean absolute error MAE are reported for three interaction strengths $g$, considering networks trained on $N = 1, 2$ and on $N = 1, 2, 3$. The test results are shown for the particle numbers $N$ included in the training set (number of training instances in parenthesis), for the extrapolations to $N = 3$ and to $N = 4$, and for the accelerated learning.

| | $g = 0.05$ | | $g = 0.26$ | | $g = 1$ | |
|---|---|---|---|---|---|---|
| | $R^2$ | MAE | $R^2$ | MAE | $R^2$ | MAE |
| Trained with $N = 1, 2$ | | | | | | |
| $N = 1$ (360000) | 0.9994 | 0.0078 | 0.9994 | 0.0076 | 0.9992 | 0.0082 |
| $N = 2$ (30000) | 0.9994 | 0.0138 | 0.9986 | 0.0231 | 0.9975 | 0.0298 |
| $N = 3$ (Extrap.) | 0.9641 | 0.1889 | 0.9335 | 0.3049 | 0.9445 | 0.2280 |
| $N = 4$ (Extrap.) | 0.8846 | 0.4241 | 0.7409 | 0.8045 | 0.8427 | 0.5164 |
| Accelerated learning for $N = 3$ | | | | | | |
| $N = 1$ (360000) | 0.9996 | 0.0063 | 0.9993 | 0.0081 | 0.9993 | 0.0081 |
| $N = 2$ (30000) | 0.9996 | 0.0123 | 0.9986 | 0.0223 | 0.9978 | 0.0277 |
| $N = 3$ (1500) | 0.9994 | 0.0219 | 0.9968 | 0.0552 | 0.9928 | 0.0798 |
| Trained with $N = 1, 2, 3$ | | | | | | |
| $N = 1$ (360000) | 0.9995 | 0.0069 | 0.9993 | 0.0081 | 0.9987 | 0.0111 |
| $N = 2$ (30000) | 0.9995 | 0.0138 | 0.9989 | 0.0212 | 0.9952 | 0.0418 |
| $N = 3$ (1200) | 0.9993 | 0.0234 | 0.9975 | 0.0534 | 0.9904 | 0.0918 |
| $N = 4$ (Extrap.) | 0.9934 | 0.1140 | 0.9890 | 0.1385 | 0.9777 | 0.1935 |
| Accelerated learning for $N = 4$ | | | | | | |
| $N = 1$ (360000) | 0.9994 | 0.0074 | 0.9994 | 0.0075 | 0.9992 | 0.0087 |
| $N = 2$ (30000) | 0.9992 | 0.0165 | 0.9987 | 0.0226 | 0.9974 | 0.0301 |
| $N = 3$ (1200) | 0.9988 | 0.0346 | 0.9974 | 0.0523 | 0.9938 | 0.0789 |
| $N = 4$ (200) | 0.9987 | 0.0438 | 0.9925 | 0.1100 | 0.9882 | 0.1459 |

## 4.3 Accelerated learning in a real-case scenario

Since the computational cost of solving many-body instances increases exponentially fast with the systems size, in practical applications of supervised learning the training sets inevitably contain significantly fewer instances for the larger particle numbers. Here, we analyse the efficiency of the accelerated learning with the typical training dataset one would encounter in a real-case scenario. Specifically, this dataset includes 360000 instances for $N = 1$, 30000 for $N = 2$, 1200 for $N = 3$, and 200 for $N = 4$. It is worth noticing that more computational time is invested in the larger particle numbers, since one expects that larger systems provide more information about the scaling of the ground-state properties with the system size. The performance of the extrapolations and of the accelerated learning is summarized in Table 3, where we report the MAE and the coefficient of determination. Interestingly, the extrapolations are significantly more accurate and more consistent than those reported in Section 4.2, which were based on fewer training instances (see Table 2). In particular, the extrapolations to $N = 4$, based on training instances for $N = 1$, $N = 2$, and $N = 3$, reach $R^2 \gtrsim 0.97$. Instead, when the training set includes only $N = 1$ and $N = 2$ instances, the extrapolations to $N = 3$ and, even more, those to $N = 4$, are less accurate, reaching coefficients of determination up to $\approx 25\%$ lower. This suggests that, only when the training set includes several system sizes and is sufficiently copious, the network can learn to accurately scale the predictions to larger particle numbers. However, a definitive assessment on the extrapolation accuracy would

require testing even larger particle numbers, which are currently out of reach for the system under investigation. Notably, we observe that including in the training set 200 instances for $N = 4$ (beyond the $N = 1$, $N = 2$, and $N = 3$ instances) is sufficient to further improve the accuracy. Again, this indicates that the network is capable of transferring the knowledge acquired on smaller system sizes, using it to drastically accelerate the learning of larger-system properties, leading to systematically accurate predictions.

## 5  Summary and conclusions

We have addressed the supervised learning of the (few) dirty boson problem, considering a specific Hamiltonian which has already been implemented in cold-atom experiments. The training and the test sets have been produced via an exact diagonalization technique, avoiding the uncontrolled approximations often employed in analogous studies on the supervised learning of quantum systems. This limited our analysis to relatively small systems, specifically, up to four bosons. These datasets are made publicly available at Ref. [38] to support future comparative studies on the supervised training of deep neural networks. Our findings indicate that the learning curve, in terms of accuracy of ground-state energy predictions versus number of training instances, is approximately universal for different particle numbers and for different interaction strengths. An interesting open question is whether a completely different neural network architecture can provide even faster learning, therefore breaking the observed universal behavior.

The artificial neural network we introduced can be trained and tested on heterogeneous datasets including instances with different particle numbers. This is achieved by combining a convolutional architecture which can address disordered fields of variable spatial extent, with an additional descriptor that explicitly represents the particle number. This descriptor is fed to the final dense layers, bypassing the convolutional part. This detail constitutes a relevant innovative aspect of our architecture. Our analysis demonstrates that this network provides accurate ground-state energy predictions, independently of the particle number, at least within the system sizes we considered. Furthermore, it allows one to attempt extrapolations to particle numbers larger than those included in the training set. The accuracy of these extrapolations is variable, and it improves when more instances and larger particle numbers are included in the training set. Notably, the learning of relatively large systems can be accelerated and made consistently accurate using heterogeneous training sets including many small-size instances and only a small amount of large-size instances. This represents the typical scenario, given the rapidly growing computational cost of solving quantum models. This strategy is somewhat analogous to the transfer learning protocols commonly employed in the field of image analysis, whereby deep neural networks pre-trained on large datasets – relevant examples are the ResNet [44] and the VGG models [45] – are then specialized on the desired classification task using much smaller samples. Here we implemented transfer learning from small to larger particle numbers using heterogeneous datasets.

In future work, it would be interesting to further explore the universality of the learning curve, considering setups with different models of disorder, interatomic potentials, geometries, or particle statistics. Furthermore, it would be important to extend our analysis to larger particle numbers, possibly in combination with different computational techniques, such as, e.g., quantum Monte Carlo simulations. Additionally, it would be interesting to consider other physical quantities, e.g., the different contributions to the ground-state energy. Also in this case, model training might be accelerated using pre-trained models in a transfer-learning strategy. As a future perspective, one can envision the use of cold-atom experiments as quantum simulators to produce the datasets required to train neural networks for computationally

intractable models. In typical experiments, the number of atoms varies due to three-body recombinations, but the deterministic preparation of few-atoms systems with well-controlled atom number has recently been achieved [46, 47]. In any case, it is convenient to use the particle number as an explicit system descriptor. This allows performing supervised learning with heterogeneous datasets obtained from different experimental setups. We argue that flexible neural-network architecture and transfer learning strategies shall play a critical role in the practical applications of cold-atom quantum simulators. Furthermore, the application of classical machine-learning techniques to study quantum systems is being developed in parallel with quantum machine-learning algorithms, which may outperform in both classical and quantum tasks [48–52].

# Acknowledgements

We acknowledge useful discussions with S. Cantori, A. Dauphin, F. Isaule and I. Morera.

**Funding information** S. P. acknowledges financial support from the FAR2018 project titled "Supervised machine learning for quantum matter and computational docking" of the University of Camerino and from the Italian MIUR under the project PRIN2017 CEnTraL 20172H2SC4. This work has been partially supported by MINECO (Spain) Grant No. FIS2017-87534-P. We acknowledge financial support from Secretaria d'Universitats i Recerca del Departament d'Empresa i Coneixement de la Generalitat de Catalunya, co-funded by the European Union Regional Development Fund within the ERDF Operational Program of Catalunya (project QuantumCat, ref. 001-P-001644). S. P. also acknowledges the CINECA award under the ISCRA initiative, for the availability of high performance computing resources and support. P. M. acknowledges funding by CAIB through the QUAREC project (PRD2018/47).

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
