# Peer review of "Supervised learning of few dirty bosons with variable particle number"

_SciPost Physics, doi:SciPost Phys. 10, 073 (2021)_

## Round 1 · Referee Report · Anonymous (Referee 1) · 2020-11-3

Strengths

The paper includes a proposal for analyzing quantum systems of cold atoms in random electromagnetic potentials, via employing neural networks. The paper seems sound and appears as important. The field of machine learning for the sciences has significantly grown in the past few years, and this article highly contributes to this area, by providing machine learning algorithms for assessing cold atom systems.

Weaknesses

In principle the paper is sound, but in my view it can be further enhanced, see below.

Report

The paper includes a proposal for analyzing quantum systems of cold atoms in random electromagnetic potentials, via employing neural networks. The paper seems sound and appears as important. The field of machine learning for the sciences has significantly grown in the past few years, and this article highly contributes to this area, by providing machine learning algorithms for assessing cold atom systems.

In my view the paper can still be further enhanced following these comments: - How does decoherence affect the training? in particular, are three body losses properly included in the formalism? I may expect they are, given that the training can be done with variable number of atoms, but this should be explicitly mentioned in a paragraph. - A further paragraph giving more details on how this algorithm may perform under scaling up should be included. How about employing, in the future, a quantum neural network to study this quantum system? - A recent reference on quantum machine learning, both with respect to quantum algorithms for machine learning, and machine learning algorithms for quantum systems, has been published in: Quantum machine learning and quantum biomimetics: A perspective, Mach. Learn.: Sci. Technol. 1 033002 (2020). https://iopscience.iop.org/article/10.1088/2632-2153/ab9803 In my view, this updated reference should be included.

I will give my final recommendation following the appropriate implementation of these suggestions.

Requested changes

In my view the paper can still be further enhanced following these comments: - How does decoherence affect the training? in particular, are three body losses properly included in the formalism? I may expect they are, given that the training can be done with variable number of atoms, but this should be explicitly mentioned in a paragraph. - A further paragraph giving more details on how this algorithm may perform under scaling up should be included. How about employing, in the future, a quantum neural network to study this quantum system? - A recent reference on quantum machine learning, both with respect to quantum algorithms for machine learning, and machine learning algorithms for quantum systems, has been published in: Quantum machine learning and quantum biomimetics: A perspective, Mach. Learn.: Sci. Technol. 1 033002 (2020). https://iopscience.iop.org/article/10.1088/2632-2153/ab9803 In my view, this updated reference should be included.

  • validity: high
  • significance: high
  • originality: good
  • clarity: high
  • formatting: excellent
  • grammar: excellent

Author:  Pere Mujal  on 2021-01-22  [id 1170]

(in reply to Report 1 on 2020-11-03)

THE REFEREE WRITES: Strengths The paper includes a proposal for analyzing quantum systems of cold atoms in random electromagnetic potentials, via employing neural networks. The paper seems sound and appears as important. The field of machine learning for the sciences has significantly grown in the past few years, and this article highly contributes to this area, by providing machine learning algorithms for assessing cold atom systems.

OUR RESPONSE: We thank the Referee for their careful reading of our manuscript and for stating that the reported results appear sound and important.

THE REFEREE WRITES: Weaknesses In principle the paper is sound, but in my view it can be further enhanced, see below.

OUR RESPONSE: We tried to accommodate the Referee’s suggestions, as explained below.

THE REFEREE WRITES: The paper includes a proposal for analyzing quantum systems of cold atoms in random electromagnetic potentials, via employing neural networks. The paper seems sound and appears as important. The field of machine learning for the sciences has significantly grown in the past few years, and this article highly contributes to this area, by providing machine learning algorithms for assessing cold atom systems.

In my view the paper can still be further enhanced following these comments:

-How does decoherence affect the training? in particular, are three body losses properly included in the formalism? I may expect they are, given that the training can be done with variable number of atoms, but this should be explicitly mentioned in a paragraph.

OUR RESPONSE: We trained our flexible neural networks on synthetic datasets obtained via exact-diagonalization computations. As stated in the conclusions, we do envision the use of cold-atom experiments as quantum simulators to produce training datasets. As the Referee correctly points out, in this setup it is essential to consider different particle numbers, since in the experiment the number of particles is not fixed due to three-body losses. Still, it is worth mentioning that the deterministic preparation of few-atom systems with controllable particle numbers has been achieved; see, e.g., F. Serwane et al., Science 332, 336-338 (2011) and Wenz et al., Science 342, 457-460 (2013). Following the Referee’s comment, in the conclusions of the revised manuscript we expand the discussion on cold-atom quantum simulators, including reference to the few-body experiments just mentioned.

THE REFEREE WRITES: - A further paragraph giving more details on how this algorithm may perform under scaling up should be included. How about employing, in the future, a quantum neural network to study this quantum system?

OUR RESPONSE: Our goal is to develop a classical neural-network model to describe quantum systems. The Referee’s suggestion to consider a quantum model, such as a quantum neural network, is quite interesting. However, it is clearly beyond the scope of our work. Following the Referee’s suggestion, in the revised manuscript we mention, in the conclusions, the perspective of using quantum models or quantum algorithms, making reference to the article pointed out by the Referee (see next comment) and to others.

THE REFEREE WRITES: - A recent reference on quantum machine learning, both with respect to quantum algorithms for machine learning, and machine learning algorithms for quantum systems, has been published in: Quantum machine learning and quantum biomimetics: A perspective, Mach. Learn.: Sci. Technol. 1 033002 (2020). https://iopscience.iop.org/article/10.1088/2632-2153/ab9803 In my view, this updated reference should be included.

OUR RESPONSE: We thank the Referee for pointing this interesting reference to us. In the revised manuscript, we make reference to this article in the conclusions, where we mention the possible future use of quantum algorithms/models to describe the quantum system under consideration.

THE REFEREE WRITES: I will give my final recommendation following the appropriate implementation of these suggestions.

OUR RESPONSE: We hope that, in view of the discussions provided above and the changes implemented in the revised manuscript, the Referee will be in the position to provide their final recommendation for publication.

---

## Round 1 · Referee Report · Anonymous (Referee 2) · 2020-12-10

Strengths

See report

Weaknesses

See report

Report

The manuscript "Supervised learning of few dirty bosons with variable particle number" by Mujal et al. reports on the application of a deep neural network to predict the ground state (GS) energy of a given speckle potential with a determined number of few interacting bosons. The neural network gets the information on the number of particles and incorporates it through a descriptor that bypasses some of the convolutional layers. The authors make three claims on the capabilities of such a network:

  1. When trained with a dataset with a mixed number of particles, it can accurately predict the GS energy for all number of particles it was trained for.

  2. When trained with a large dataset for 1...N particles, and a small dataset for N+1 particles, the network accurately predicts the GS energy of N+1 particles. The authors call this scenario "transfer learning".

  3. When the network is trained on a dataset for 1...N particles, it will be able to predict the GS energy for N+1 particles reasonably well.

The study is based on numerical calculations done with up to N=4 particles. Overall, I find that the first two claims are convincing, while the third one is not. Looking at figures 3 and 4, it is clear that there is a clear systematic deviation between the extrapolation (red points) and the ground truth. Moreover, this deviation depends (as I would expect) on the interaction parameter. However, I do think that claims 1 and 2 by themselves merit the publication of this paper as they open a new pathway to improve the accuracy and reduce the computational cost of numerical simulations. Regarding claim 3, with current data, I think the authors should either remove it or better support it (see also points 1-3 below). I list several points the authors should address before final acceptance.

  1. To better support the extrapolation claim, I would suggest the authors to consider the case where the network is trained for 1...N particles and extrapolation is made to the N+2 case. With their current data, this can be done with N=1,2 and comparing the extrapolation to N=4. This check can give a sense of how fast the correlation of the GS energy with N decays.

  2. The authors do not relate their working points (in terms of interaction and disorder strength) to known phases of the dirty boson model (e.g., Phys. Rev. Lett. 98, 170403, and Phys. Rev. B 80, 104515). I would expect that the scaling with N of the GS energy in different phases would be very different. I believe the paper could be improved by making this connection.

  3. It could be interesting to calculate for each numerical point not only the total energy, but also the kinetic, interaction, and potential energies separately, and then see if the deviation in the extrapolation is related to an increase in one of them in particular.

  4. Regarding the universal scaling in training, did the authors check how the exponent depends on the Keras learning rate parameter? could the so-called universal scaling be a result of using the same training parameters?

  5. In figures 3 and 4 it would make more sense to subtract the trivial linear dependence, thus giving a better characterization of the prediction deviation.

  6. The reported training did not perform regularization, which is claimed to contribute only a slight improvement. This is rather unusual in the realm of deep learning. Can the authors explain how the model evades over-fitting at local minima? Do they believe that this particular problem is characterized by a convex parameter hyperspace?

  7. At the end of section 4.1, the authors write, "These results indicate that the combined training with smaller sizes provides a boost to the learning process for the larger size, allowing the network to reach high accuracy with fewer training instances." It could be helpful to compare this result to the case where the 200 realizations are augmented before training. Note also that the usage of the word "augmented" (page 3) may be confusing in the machine-learning community, where it is usually attributed to synthetic manipulation of the input data to extend a given data set (see Ref. 39).

Requested changes

See report

  • validity: good
  • significance: good
  • originality: good
  • clarity: high
  • formatting: good
  • grammar: excellent

Author:  Pere Mujal  on 2021-01-22  [id 1169]

(in reply to Report 2 on 2020-12-10)

THE REFEREE WRITES: The manuscript "Supervised learning of few dirty bosons with variable particle number" by Mujal et al. reports on the application of a deep neural network to predict the ground state (GS) energy of a given speckle potential with a determined number of few interacting bosons. The neural network gets the information on the number of particles and incorporates it through a descriptor that bypasses some of the convolutional layers. The authors make three claims on the capabilities of such a network: 1. When trained with a dataset with a mixed number of particles, it can accurately predict the GS energy for all number of particles it was trained for. 2. When trained with a large dataset for 1...N particles, and a small dataset for N+1 particles, the network accurately predicts the GS energy of N+1 particles. The authors call this scenario "transfer learning". 3. When the network is trained on a dataset for 1...N particles, it will be able to predict the GS energy for N+1 particles reasonably well. The study is based on numerical calculations done with up to N=4 particles. Overall, I find that the first two claims are convincing, while the third one is not. Looking at figures 3 and 4, it is clear that there is a clear systematic deviation between the extrapolation (red points) and the ground truth. Moreover, this deviation depends (as I would expect) on the interaction parameter. However, I do think that claims 1 and 2 by themselves merit the publication of this paper as they open a new pathway to improve the accuracy and reduce the computational cost of numerical simulations. Regarding claim 3, with current data, I think the authors should either remove it or better support it (see also points 1-3 below). I list several points the authors should address before final acceptance.

OUR RESPONSE: We thank the Referee for their careful reading, and for stating that claims 1 and 2 merit publication. We emphasise that we did not intend to convey the message that the extrapolations are sufficiently accurate. Our main message is that the variable-N neural network allows us to implement an accelerated learning procedure, whereby the learning of relatively large systems is accelerated using data for smaller system sizes. In the revised manuscript, we scale down or rephrase certain possibly misleading sentences. Still, here it is worth mentioning that in the case of the predictions to N=4 (from training with N=1, 2, and 3) the results are not so inaccurate; furthermore, in the real-case scenario discussed in Section 4.3, the extrapolation accuracy reaches a coefficient of determination of R^2>0.97. These results led us to describe, in the previous version of the manuscript, the extrapolations as “fairly accurate”. Anyway, in the revised manuscript we emphasise that the extrapolation accuracy is not consistently accurate for practical applications, and we only speculate that they might become reliable if even larger particle-numbers are included in the training set.

THE REFEREE WRITES: 1. To better support the extrapolation claim, I would suggest the authors to consider the case where the network is trained for 1...N particles and extrapolation is made to the N+2 case. With their current data, this can be done with N=1,2 and comparing the extrapolation to N=4. This check can give a sense of how fast the correlation of the GS energy with N decays.

OUR RESPONSE: As discussed in the previous reply, we do not intend to convey the message that the extrapolations are sufficiently accurate. In the revised manuscript, we have removed or rephrased all sentences that might mislead to this conclusion, and we better emphasize that our main message is on the possibility of performing accelerated learning, discussing the performance of this procedure.

THE REFEREE WRITES: 2. The authors do not relate their working points (in terms of interaction and disorder strength) to known phases of the dirty boson model (e.g., Phys. Rev. Lett. 98, 170403, and Phys. Rev. B 80, 104515). I would expect that the scaling with N of the GS energy in different phases would be very different. I believe the paper could be improved by making this connection.

OUR RESPONSE: We address a few-body system. We do consider different regimes of the interaction strengths, including weak, intermediate, and also strong interaction close to the Tonks-Girardeau limit. As stated in the manuscript, we observe the same learning speed in all regimes. Due to the small systems size, it is not possible to identify the different phases discussed in the articles mentioned by the Referee. Still, we think that those references represent relevant articles on the dirty boson problem, and in the revised manuscript we make reference to those articles when we mention the dirty boson problem.

THE REFEREE WRITES: 3. It could be interesting to calculate for each numerical point not only the total energy, but also the kinetic, interaction, and potential energies separately, and then see if the deviation in the extrapolation is related to an increase in one of them in particular.

OUR RESPONSE: In this manuscript we analyse predictions of ground-state energies of the addressed quantum systems. This is a relevant quantity. For example, in quantum chemistry it allows for the identification of the equilibrium molecular configuration, while in molecular-dynamics it allows extracting force fields. The suggestions made by the Referee are indeed quite interesting. However, analysing different physical quantities is beyond the scope of this work. Since we intend to investigate different observables in future work, in the revised manuscript we mention this possibility in the Conclusions section.

THE REFEREE WRITES: 4. Regarding the universal scaling in training, did the authors check how the exponent depends on the Keras learning rate parameter? could the so-called universal scaling be a result of using the same training parameters?

OUR RESPONSE: We trained all neural networks using the ADAM algorithm with default parameters. We considered different stopping criteria to halt the training process. The apparently universal learning speed appears to be independent of these details. However, we emphasize here (as already stated in the manuscript) that the approximate universality is related to different particle numbers (both with homogeneous and with heterogeneous training sets) and to different interaction strengths. An interesting open question is whether a completely different neural network architecture can provide even faster learning, therefore breaking the observed universal behavior. Following the Referee’s comment, we mention this possibility in the revised manuscript.

THE REFEREE WRITES: 5. In figures 3 and 4 it would make more sense to subtract the trivial linear dependence, thus giving a better characterization of the prediction deviation.

OUR RESPONSE: This is a useful suggestion. However, we prefer to adhere to a common practice in this field to simply visualize predicted versus ground-truth values [see, e.g., Phys. Rev. A 96, 042113 (2017); Chem. Sci. 10, 4129 (2019); ChemSystemsChem 2, e1900052 (2020)]. To better characterize the prediction deviations we have added the right panels on Figs. 3 and 4, where we show the distributions of the absolute error.

THE REFEREE WRITES: 6. The reported training did not perform regularization, which is claimed to contribute only a slight improvement. This is rather unusual in the realm of deep learning. Can the authors explain how the model evades over-fitting at local minima? Do they believe that this particular problem is characterized by a convex parameter hyperspace?

OUR RESPONSE: The possibility of incurring in the over-fitting problem is usually related to the number of training instances available. When the model is trained with several thousand instances, the risk of over-fitting is reduced. It is also worth mentioning that over-fitting is not always related to reaching a local minimum. In fact, the absolute minimum might in fact over-fit the training data. We do not believe that the optimization problem is characterized by a convex landscape. Furthermore, neural networks have proven remarkable generalization performances in many applications, outperforming other universal-function approximators. In the cases with relatively fewer training instances, we inspect for the occurrence of over-fitting by comparing the MAE of the test set against the MAE of the training set. In general, we find comparable results (in the worst case, we get MAE_test~2MAE_train with 200 instances), indicating that the residual prediction error is not dominated by the overfitting problem. We observe that tuning the regularization parameter (with L2 regularization) provides marginal improvements, and only for the smallest training sets we consider. In the revised manuscript, we expand the discussion on the overfitting problem in Section 2, describing in more detail how we inspect for the occurrence of overfitting and providing some quantitative measures.

THE REFEREE WRITES: 7. At the end of section 4.1, the authors write, "These results indicate that the combined training with smaller sizes provides a boost to the learning process for the larger size, allowing the network to reach high accuracy with fewer training instances." It could be helpful to compare this result to the case where the 200 realizations are augmented before training.

OUR RESPONSE: We have to clarify that, when we use the “augmented” training set with many small-N instances and the few large-N instances, we do perform the training from scratch. Indeed, there is no benefit here in performing the training in two stages (e.g., training first on N=1,2, and 3 instances, and the re-train on N=4), since we do have access to the whole dataset. This scenario is somewhat different compared to the one typically encountered in the field of image analysis, whereby deep networks pretrained on large datasets (usually not available to the final user) are specialized on the available (smaller) datasets in a separate process. In the revised manuscript, we provide a more explicit description of the accelerated training process.

THE REFEREE WRITES: Note also that the usage of the word "augmented" (page 3) may be confusing in the machine-learning community, where it is usually attributed to synthetic manipulation of the input data to extend a given data set (see Ref. 39).

OUR RESPONSE: To avoid confusion, in the revised manuscript we replace the possibly misleading word “augmented” with “merged”.

---

## Round 2 · Referee Report · Anonymous (Referee 1) · 2021-1-26

Report

The authors have properly enhanced the manuscript following my remarks. I can recommend it for publication in current form.
  • validity: high
  • significance: high
  • originality: high
  • clarity: high
  • formatting: good
  • grammar: good

Author:  Pere Mujal  on 2021-02-28  [id 1271]

(in reply to Report 1 on 2021-01-26)

THE REFEREE WRITES:
The authors have properly enhanced the manuscript following my remarks. I can recommend it for publication in current form.
OUR ANSWER:
We thank the Referee for the positive assessment on our manuscript and for recommending its publication.

---

## Round 2 · Referee Report · Anonymous (Referee 2) · 2021-2-15

Strengths

see repot

Weaknesses

see report

Report

Although the authors have addressed all of my comments, their response is rather disappointing. Most of the changes made are in more careful wording of the claims. Of course, this is welcome, but my suggestions, most of which declined politely by the authors, were meant to provide the reader with a better understanding of the result. For example, point 1 in my original review could have shown the limitations of extrapolation and could easily have been done with their already existing data. Unfortunately, the authors did not accept my suggestion. The same goes for points 2,3 and 4. In point 5, the authors again do not take my suggestion, but instead, present a histogram of the absolute error. This addition is not helpful, in my opinion. My original proposal was to subtract the linear term and still plot the data as a function of E. That way, one could see the systematic deviation with the energy. The authors choose not to do that; I suspect because the result is not favorable. My overall feeling is that the authors decided to make only small "cosmetic" changes. However, as already noted in my first review, the paper's two established claims justify its publication, and since in this version the authors softened the third claim, I can recommend its acceptance.
  • validity: good
  • significance: good
  • originality: good
  • clarity: high
  • formatting: good
  • grammar: excellent

Author:  Pere Mujal  on 2021-02-28  [id 1272]

(in reply to Report 2 on 2021-02-15)

THE REFEREE WRITES:
Although the authors have addressed all of my comments, their response is rather disappointing. Most of the changes made are in more careful wording of the claims. Of course, this is welcome, but my suggestions, most of which declined politely by the authors, were meant to provide the reader with a better understanding of the result. For example, point 1 in my original review could have shown the limitations of extrapolation and could easily have been done with their already existing data. Unfortunately, the authors did not accept my suggestion.
OUR RESPONSE:
As requested by the Referee, in this second revision of the manuscript we include data and comments on the extrapolation from N=1 and N=2 instances to the particle number N=4. This analysis is reported for the real-case scenario discussed in Section 4.3 and in Table 3. As expected, the extrapolation accuracy is reduced compared to the extrapolation from N=1,2,3, by approximately 10%-20%. While this analysis further highlights the limitations of the extrapolations procedure and the need to use sufficiently large sizes in the training stage (as, in fact, already discussed in the previous version of the manuscript), it does not affect our two main claims. We hope that the additional data and the further discussions on the limitation of the extrapolation procedure can be considered an adequate response to the main concern raised by the Referee.

THE REFEREE WRITES:
The same goes for points 2,3 and 4.
OUR RESPONSE:
It seems to us that these three points have been given appropriate consideration, at least within the framework set by the model Hamiltonian we address.
Concerning point 2: we do state that we observe the same approximately universal behavior in all physical regimes, from weakly interacting to the Tonks-Girardeau limit. Phases such as the Bose glass cannot be identified in the few-body model we consider.
Concerning point 3: we mention the possibility to consider different physical quantities. However, this analysis is clearly beyond the scope of our work, and it wouldn't necessarily add significant information concerning the two main claims we report on.
Concerning point 4: we do state that the learning curve is not affected by training parameters such as the learning rate.

THE REFEREE WRITES:
In point 5, the authors again do not take my suggestion, but instead, present a histogram of the absolute error. This addition is not helpful, in my opinion. My original proposal was to subtract the linear term and still plot the data as a function of E. That way, one could see the systematic deviation with the energy. The authors choose not to do that; I suspect because the result is not favorable.
OUR RESPONSE:
We believe that a histogram of the discrepancies is a legitimate and effective way to visualize the magnitude of these discrepancies. One can directly read the value of the maximum discrepancy, and also have quantitative information on the probability to find a given value. Indeed, the importance of using accelerated learning instead of simple extrapolations is quite evident from these figures. As the Referee implies, the information on the energy dependence of the discrepancy is not visible in the histograms. However, this information is provided by the scatter plot of predicted versus exact energies. For full transparency, we make use of the SciPost feature of making authors' responses publicly available, attaching here the plots showing the data as suggested by the Referee, i.e., plotting the discrepancy between predicted and exact energies, versus exact energies. It seems to us that the plots reported in the manuscript provide readers all information necessary to justify our claims, and also to understand the limitations of the extrapolation procedure.

THE REFEREE WRITES:
My overall feeling is that the authors decided to make only small "cosmetic" changes. However, as already noted in my first review, the paper's two established claims justify its publication, and since in this version the authors softened the third claim, I can recommend its acceptance."
OUR RESPONSE:
We hope that the additional data included in the manuscript, the (publicly available) graphs included in this response, and the above discussions, will be considered an adequate response to the Referee's concerns. The additional information further highlights the limitations of the extrapolation procedure, and the need to use sufficiently copious training sets including several system sizes. The necessity of exploring even larger sizes to establish the applicability of direct extrapolations was already emphasized in the manuscript. Since the Referee already agreed that the two main claims -- i.e., the implementation of a flexible neural network for variable particle numbers and the efficiency of accelerated learning -- had already been established, we hope that the Referee will find that his/her suggestions have been exhaustively addressed.

Attachment:

Figures_Response.pdf

---

## Round 2 · List of Changes

1. We have reformulated the abstract, following the second Referee’s comment about the accuracy of the extrapolations to larger system sizes.
    1. In the introduction, we have added citations to new Refs. [29] and [30], which were mentioned by the second Referee in his/her report.
    2. We have modified the introduction according to the second Referee’s report comments on the extrapolations. The claim on the extrapolation accuracy is substantially scaled down.
    3. At the end of Sec. 2.3, we have extended the discussion on the use of regularization techniques and on the procedure we adopted to inspect for the possible occurrence of overfitting (see comment by the second Referee).
    4. In Secs. 4.1, 4.2, and 4.3, we have modified the claims and discussion about the extrapolations, emphasising the specific conditions where reasonable accuracy is obtained, and mentioning the need for further analysis on larger systems.
    5. We have added the right panels in Figs. 3 and 4 to better visualize the discrepancies in the extrapolations and in the outcomes of accelerated learning (see comment by the second Referee).
    6. In the conclusions, we have added a sentence about the approximately universal behaviour of the learning curve, speculating that different neural network architecture might provide a faster learning (see comment by the second Referee).
    7. In the conclusions, we have mentioned the possibility of computing other physical quantities (see comment by the second Referee).
    8. In the conclusions, we expand the discussion on cold-atom quantum simulators and on three-body recombinations (see comment by the first Referee). We cite new Refs.[46] and [47], which report cold-atom experiments on the deterministic preparation of few-body systems with controlled atom numbers.
    9. In the conclusions, we have pointed out the possibility of using quantum machine learning, citing new Refs. [48-52], following a comment by the first Referee.

---

## Round 3 · List of Changes

- In Table 3, we include additional data for the extrapolation from the system sizes N=1 and N=2, to N=4.
- In Section 4.3, we discuss the additional data concerning the extrapolations from N=1 and N=2 to N=4. The limitations of the extrapolation procedure are further emphasized, highlighting the need to use several system sizes in the training set. The need of a further investigation on the accuracy of the extrapolation procedure was already stressed.
- In the "Summary and conclusions" Section 5, the possible inaccuracies of the direct extrapolations are more clearly highlighted, mentioning the improvements obtained when larger system sizes are included in the training set.

---

## Editorial Decision

published